# High Prevalence of Multidrug-Resistant, Biofilm-Forming Virulent *Clostridium perfringens* in Broiler Chicken Retail Points in Northeast India

**DOI:** 10.3390/foods12224185

**Published:** 2023-11-20

**Authors:** Govindarajan Bhuvana Priya, Kandhan Srinivas, Heiborkie Shilla, Arockiasamy Arun Prince Milton

**Affiliations:** 1College of Agriculture, Central Agricultural University (Imphal), Kyrdemkulai 793105, Meghalaya, India; bhuvana.priya20@gmail.com; 2Division of Animal and Fisheries Sciences, ICAR Research Complex for NEH Region, Umiam 793103, Meghalaya, India; jsrinivaskandan@gmail.com (K.S.); heiborkie@gmail.com (H.S.)

**Keywords:** *Clostridium perfringens*, broilers, retail points, biofilm-forming ability, multidrug resistance, India

## Abstract

In light of the significant public health and food safety implications associated with *Clostridium perfringens*, this study aimed to isolate and characterize *C. perfringens* in samples obtained from broiler chicken retail points in Meghalaya, northeastern India. A total of 280 samples comprising meat, intestinal contents, water, and hand swabs were processed to detect contamination by *C. perfringens*. The isolates were subjected to toxinotyping, antimicrobial susceptibility testing, and biofilm-forming ability test. The overall occurrence of *C. perfringens* was 22.5% (17.74–27.85, 95% CI) with the highest recovery from intestine samples (31%; 22.13–41.03, 95% CI), followed by meat (23%, 15.17–32.49, 95% CI) and water samples (18%, 8.58–31.44, 95% CI). Type A was the predominant toxinotype (71.43%, 58.65–82.11, 95% CI), followed by Type A with beta2 toxin (17.46%, 9.05–29.10, 95% CI), Type C (7.94%, 2.63–17.56, 95% CI), and Type C with beta2 toxin (3.17%, 0.39–11.0, 95% CI). Nearly all (95.24%) isolates were multidrug resistant and 68.25% were biofilm formers. The predominance of multidrug-resistant and virulent Type A and Type C *C. perfringens* in retail broiler meat and intestines in the tribal-dominated northeastern region of India is of great concern from food safety and public health perspectives.

## 1. Introduction

*Clostridium perfringens* is a food-borne biological hazard of bacterial origin; they are Gram-positive rods thriving on anaerobic respiration and with spore-forming ability [1]. In addition to being an ubiquitous member of the environmental microflora, *C. perfringens* has the potential to cause diseases in various species, including poultry and humans [1]. The poultry industry is often marred by outbreaks of necrotic enteritis opportunistically caused by *C. perfringens*, resulting in severe economic losses. Colonization in poultry occurs early in life, even during the hatchery stage, with potential to disseminate in the broiler value chain [2]. The predominance of pathogenic strains of *C. perfringens* expressing various virulence factors over non-pathogenic strains triggers necrotic enteritis. In humans, the organism is commonly known for its food-borne importance amidst an array of other clinical manifestations [1]. In that regard, *C. perfringens* is the predominant bacterial and second-most common cause of food-borne illnesses in Canada, next to norovirus [3]. World Health Organization estimates in 2010 placed *C. perfringens* as the leading causal agent among food-borne intoxications, with 3,998,164 foodborne illnesses [4].

The virulence arsenal of *C. perfringens* comprises various toxin genes mediated by plasmids as well as chromosomes. Strains of *C. perfringens* are classified into seven toxinotypes (A, B, C, D, E, F and G) based on occurrence patterns of toxin genes (α, β, ε, ι, *cpe* and *net*B) [5]. The *cpa* gene, present on the variable sections of the chromosome, encodes the phospholipase C (α-toxin) and is identifiable in all toxinotypes [5,6]. The *C. perfringens* enterotoxin (CPE), often associated with food poisoning in humans, is either chromosome- or plasmid-mediated [1]. The CPE is usually present in marginal proportions of all *C. perfringens* strains, with higher occurrence in Type A strains, and is associated with bowel disturbances in humans and animals [7]. The beta toxin (*cpb2*) was first identified in the case of a Type C-infected piglet with haemorrhagic necrotic enteritis [8]. The beta2 toxin, capable of being produced by all toxinotypes, is considered a lethal and necrotizing factor that contributes to gastro-intestinal derangements in humans as well as animals [8]. Recently, a large clostridial toxin named *TpeL* was identified in Type C *C. perfringens* and was an important contributor to necrotic enteritis [9].

The presence of antimicrobial-resistant strains of bacteria in the food chain has been a constant public health concern. Antibiotic resistance in *C. perfringens* is chiefly mediated by plasmids, as well as transposons and insertion sequences [10]. Increasing reports of *C. perfringens* resistance to tetracycline, lincomycin, and erythromycin have been a growing concern over recent years, made worse by the ability of *C. perfringens* to produce biofilm and form spores [11]. Biofilms enhance resistance to antimicrobials and disinfectants and are associated with increased virulence and pathogenesis [12]. The molecular interplay of genes resulting in biofilm formation by *C. perfringens* was only recently elucidated [13]. The ability to produce spores, secrete toxins, form biofilms, and harbour antimicrobial resistance genes, coupled with a short (<10 min) doubling time, makes *C. perfringens* an important threat to humans and animals.

*C. perfringens* has been widely reported in various parts of India [1,14,15]. However, except for a case study of *C. perfringens* involving six birds on a farm, there are no other systematic reports of prevalence, virulence, and antimicrobial resistance of *C. perfringens* in the broiler chicken value chain in Meghalaya, a northeast Indian state with a dominant tribal population [16]. Interestingly, chicken intestines have culinary value regionally. Our objective was to determine the food-borne threat posed by *C. perfringens*.

## 2. Materials and Methods

### 2.1. Sample Collection and Isolation

A total of 280 samples comprising broiler chicken meat (n = 100), chicken intestine (n = 100), slaughter water (n = 50), and butcher hand swabs (n = 30) were collected from retail chicken meat shops in 2 districts (East Khasi Hills and Ri-Bhoi) of Meghalaya, India. Hand swabs were collected during meat cutting, and slaughter water refers to the water used for slaughtering and meat cutting at the retail points. Within 2 h of collection, samples were inoculated in 10 mL of Robertson cooked meat (RCM) broth (HiMedia Laboratories, Mumbai, India). After incubation at 37 °C for 24 h, the enriched inoculum was plated onto 5% sheep blood agar plates (Figure 1) that were incubated at 37 °C for 24 h under an anaerobic environment induced with a gas pack system (Anaerogen™ 2.5 L; Thermo Scientific, Hampshire, UK). Presumptive identification of *C. perfringens* was attempted by exploiting the ability of the organism to produce a double zone of haemolysis on the sheep blood agar; suspected colonies were propagated in RCM broth. Further confirmation was achieved by PCR-based detection of *cpa* (α-toxin) using *C. perfringens* ATCC 13124 as a positive reference. Confirmed isolates were cryopreserved in brain heart infusion broth containing 15% glycerol and stocked at −80 °C.

### 2.2. Toxinotyping of Confirmed Isolates

For all isolates, genomic DNA was extracted with a QIAmp DNA Mini Kit (Qiagen, Hilden, Germany) using the manufacturer’s protocol. Then, PCR-based toxinotyping was undertaken, as described, by screening for *cpb*, *cpe*, *etx*, *iap*, *cpb2*, and *net*B [17,18]. Additionally, isolates were screened for *tpeL* that codes for a large clostridial toxin [9]. Positive controls for various toxin genes were obtained from our previous studies [1,14]. Thermocycling was performed using an Eppendorf Master cycler^®^ thermal cycler (Eppendorf, Hamburg, Germany), with PCR products separated on 1.5% agarose gels, stained with ethidium bromide, and amplicons visualized with UV illumination. All procedures were conducted twice to ensure consistency.

### 2.3. Antimicrobial Susceptibility Testing and MAR Indexing

The disk diffusion method (Kirby Bauer) was used to determine the antibiogram profile of *C. perfringens* isolates against antibiotics approved for clinical use. Erythromycin (ERY, 15 µg), clarithromycin (CLR, 15 µg), ampicillin (AMP, 10 µg), chloramphenicol (CHL, 30 µg), clindamycin (CLI, 2 µg), linezolid (LZD, 30 µg), ofloxacin (OFX, 5 µg), penicillin (PEN, 10 U), co-trimoxazole (STX, 25 µg), tetracycline (TET, 30 µg), and azithromycin (AZM, 15 µg) were placed on the lawn-inoculated plates of Mueller–Hinton agar, and incubated anaerobically at 37 °C for 24 h. Interpretative criteria for *Staphylococcus aureus* were adopted, as breakpoints for *C. perfringens* were unavailable [19]. All procedures were performed twice. Isolates resistant to 3 or more classes of antimicrobials were designated as multidrug resistant. Multiple antibiotic resistance (MAR) index of the isolates was calculated as the ratio of number of antibiotics towards which resistance is observed to the number of antibiotics exposed.

### 2.4. Evaluation of Biofilm-Forming Ability

A crystal violet-based assay was used to evaluate the biofilm-forming ability of *C. perfringens* isolates. First, pure isolates were propagated in tryptone soy broth (TSB) (Himedia Laboratories, Mumbai, India) anaerobically at 37 °C for 24 h. Further, the broth inoculum was diluted in fresh TSB and subsequently aliquoted to a 96-well polystyrene plate (Nunc™, Fisher Scientific, Waltham, MA, USA). After incubating the plate at 37 °C in an anaerobic environment for 24 h, the planktonic suspension was removed with the help of 1% phosphate-buffered saline (PBS) wash and wells were stained with crystal violet (1%) solution (SRL, Mumbai, India). After incubating at room temperature (30 min), wells were washed with 1% PBS. Ethanol was added to the stained wells and incubated for 15 min. Absorbance was read at 595 nm with a microplate reader (NanoQuant infinite M200PRO, Tecan, Männedorf, Switzerland) and biofilm-forming ability was interpreted as described [1].

### 2.5. Hierarchical Clustering, Heatmap, and Correlation Plot Analysis

For all isolates, data were transformed into binary variables and used for construction of a heatmap and correlation plot analyses. Heatmaps with hierarchical clustering were constructed using “pheatmap” and “dendextend” (R software version 4.0.5) packages. Correlation plots with Spearman’s rank correlation method were made after converting data into a correlation matrix using the “corrplot” in R software version 4.0.5.

### 2.6. Statistical Analysis

Fisher’s exact test was performed with the help of MS-Excel to analyse the statistical association between various variables. For all the proportions, confidence intervals (CI) were calculated with 95% confidence level using binomial exact calculation (https://sample-size.net/confidence-interval-proportion/ accessed on 2 July 2023).

## 3. Results

### 3.1. Occurrence of C. perfringens

The overall occurrence of *C. perfringens* in retail chicken-associated samples was 22.5% (17.74–27.85 95% CI; 63/280) with the highest recovery from intestine samples (31%, 22.13–41.03 95% CI; 31/100), followed by meat samples (23%, 15.17–32.49 95% CI; 23/100) and water samples (18%, 8.58–31.44 95% CI; 9/50). However, none of the hand swab samples were positive for *C. perfringens*.

### 3.2. Molecular Toxinotyping

Molecular toxinotyping (Figure 2) of the 63 isolates by screening for toxin/virulence genes (*cpa*, *cpb*, *etx*, *iap*, *cpb2*, *cpe*, *Net*B, and *tpeL*) revealed the predominance of Type A toxinotype (45/63, 71.43%; 58.65–82.11 95% CI), followed by Type A with beta2 toxin (11/63, 17.46%; 9.05–29.10 95% CI), Type C (5/63, 7.94%; 2.63–17.56 95% CI) and Type C with beta2 toxin (2/63, 3.17%; 0.39–11.0 95% CI). The most common toxin/virulence gene was *cpa* (63/63, 100%; 94.31–100 95% CI), followed by *cpb2* (13/63, 20.63%; 11.47–32.70 95% CI) and *cpb* (7/63, 11.11%; 4.59–21.56 95% CI). However, no signatures were detected among isolates for *cpe*, *etx*, *iap*, or *net*B; therefore, toxinotypes B, D, E, F and G were not present.

### 3.3. Antimicrobial Resistance Profiling and MAR Index

Regarding antimicrobial susceptibility testing (Table 1), the highest resistance was found in linezolid (61/63, 96.83%; 89–99.61 95% CI) followed by clarithromycin (58/63, 92.06%; 82.44–97.37 95% CI), erythromycin (56/63, 88.89%, 78.44–95.41 95% CI), clindamycin (55/63, 87.30%; 76.50–94.35 95% CI), azithromycin (51/63, 80.95%; 69.09–89.75 95% CI), ampicillin (45/63, 71.43; 58.65–82.11 95% CI), co-trimoxazole (26/63, 41.27%; 29.01–54.38 95% CI), tetracycline (24/63, 38.1%; 26.14–51.20 95% CI), penicillin (21/63, 33.33%; 21.95–46.34 95% CI), and chloramphenicol (18/63, 28.57; 17.89–41.35% 95% CI). All isolates were susceptible to ofloxacin (fluoroquinolone). Multidrug resistance was evaluated based on the criteria of resistance against three or more antimicrobial classes. In the present study, 60 of 63 isolates were multidrug resistant (95.24%; 86.71–99.01 95% CI). The most common resistance profiles were ERY-CLR-AMP-CLI-LZD-STX-AZM with the modal frequency (n = 6), followed by ERY-CLR-AMP-CLI-LZD-AZM and ERY-CLR-AMP-CLI-LZD-PEN-STX-AZM (n = 4). The MAR indices of the isolates ranged from 0.18 to 0.91 with a mean of 0.60. In terms of the isolation source, the mean MAR index values were 0.57, 0.59, and 0.72 for isolates from meat, intestine, and water, respectively (*p* > 0.5). Isolate S9_H from chicken intestine and isolates CP1_A and CP1_B recovered from water samples were resistant to 10 of 11 antimicrobials screened with a MAR index of 0.91. With the exception of isolate L3_E, the MAR indices of all other isolates were greater than 0.2.

### 3.4. Evaluation of Biofilm-Forming Ability

Regarding biofilm-forming ability, 43 of 63 isolates (68.25%, 55.31–79.42, 95% CI) were biofilm formers with a mean absorbance of 0.4010. Furthermore, 4, 22 and 17 isolates were deemed strong, medium, and weak biofilm formers, respectively. Mean absorbance of strong biofilm formers was 0.7217 (range: 0.6394 to 0.8746), whereas mean absorbance values of moderate and weak biofilm formers were 0.4388 (range: 0.1946 to 0.6229) and 0.2766 (range: 0.1874 to 0.3455), respectively. Isolate RUC4_2 recovered from chicken meat had the highest observed OD value. However, Fisher’s exact test could not identify any statistical association between biofilm-forming ability and other variables.

### 3.5. Heatmap-Based Hierarchical Clustering and Correlation Analyses

Heatmap construction with hierarchical clustering grouped isolates into four clusters (two large and two small clusters) (Figure 3). Cluster 1 comprised isolates belonging to Type A toxinotype, susceptible to co-trimoxazole, tetracycline, chloramphenicol, and ofloxacin. Source-wise clustering of isolates was observed (all members were isolated from the carcass). Cluster 2 was small and encompassed isolates resistant to azithromycin and tetracycline. However, all members of this small cluster were susceptible to ofloxacin. The large cluster (Cluster 3) represented isolates negative for *cpb2* and that were phenotypically susceptible to chloramphenicol, ofloxacin, and penicillin. Source-based sub-clustering was appreciable within this cluster. The members of the other large cluster (4) were predominantly Type A toxinotype with increased susceptibility to ofloxacin. Correlation plot analysis with Spearman’s correlation coefficient (Figure 4) revealed high positive correlations between various variables. In the correlation plot, the colour and size of the circles represent the magnitude and direction of correlation. There were higher levels of positive correlation (ρ > 0.9) between resistance to macrolides such as erythromycin, clarithromycin, and azithromycin and resistance to ampicillin and linezolid. Biofilm-forming ability was highly associated with linezolid resistance.

## 4. Discussion

*C. perfringens*, a ubiquitous member of the gut microflora of normal poultry, is associated with consumption of under-processed protein-rich foods such as meat, which provides a suitable environment for this organism [20]. Contamination of chicken meat by *C. perfringens* is not uncommon [20].

In the present study, we identified multidrug-resistant, biofilm-producing *C. perfringens* isolates that harboured toxin genes such as *cpa*, *cpb*, and *cpb2* in retail chicken-associated samples. In total, 22.5% of the 280 samples screened were positive for *C. perfringens*, with occurrence rates of 31, 23, and 18% in intestine, meat and water samples, respectively. Faecal prevalence of *C. perfringens* has been well established, with occurrence levels of *C. perfringens* in poultry intestinal contents ranging from 9.9 to 95%, with the wide range attributed to variable feeding and management conditions [2,21,22]. Furthermore, the use of antimicrobial growth promoters influences the level of *C. perfringens* in the gut microflora of poultry [2]. The presence of *C. perfringens* in intestinal samples in retail shops is of importance as chicken intestines have culinary value in various parts of the world including India and cross-contamination of other meat portions is another possibility.

Higher levels of *C. perfringens* in poultry meat, ranging from 18 to 88%, have been reported [20]. Differences in methodologies to isolate *C. perfringens* could explain the dispersion in the occurrence values in meat. Furthermore, the occurrence of *C. perfringens* in water samples (18%) from the slaughter area indicated that *C. perfringens* contaminated the slaughter environment, consistent with a study in a Chinese chicken production chain [23]. In that study, *C. perfringens* was also isolated from operators’ gloves, which contrasted with present results. The predominance of Type A toxinotype in this study has been validated in various parts of the world and in a variety of sources [11,14,24].

The presence of the *cpb2* gene in 20.63% of isolates raises concerns of a potential threat to food safety, as this gene is implicated with aggravating gastro-intestinal symptoms in clinical cases linked to antibiotic-associated and sporadic diarrhoea [1]. Moreover, beta toxin is also considered the chief virulence factor in Type C cases [14]. The absence of other toxin genes such as *etx, iap, cpe, netB*, and *tpeL* in *C. perfringens* have been reported [1,24,25]. The presence of various toxin genes on the plasmids offers a wide diversity of pathovars of *C. perfringens* and allows for conversion of toxinotype by uptake or loss of plasmids [6].

The indiscriminate use of antibiotics in intensively reared animals such as poultry and pigs has resulted in the wide prevalence of multidrug-resistant strains. In the present study, there were very high resistance rates for linezolid (96.83%), clindamycin (87.30%), and macrolides such as erythromycin (92.06%), clarithromycin (88.89%), and azithromycin (80.95%). Reports of linezolid resistance in Gram-positive anaerobic bacteria have been steadily surfacing in recent years [26]. Similar levels of erythromycin resistance among *C. perfringens* were identified in an earlier report from Iran investigating raw beef [11]. Moreover, a study from Egypt reported broiler-origin *C. perfringens* with 100% resistance to macrolides and lincosamides [27]. Multidrug resistance has been a glaring trend among *C. perfringens* in recent years. Alarming levels of multidrug resistance were encountered in the present study (95.25%). Various researchers have hinted at the prevalence of multidrug-resistant strains of *C. perfringens* isolated from poultry origin [23,24,27]. The irrational and unregulated use of antibiotics at veterinary care and farm levels are regarded as probable sources of multidrug-resistant strains [28].

Biofilms are a food safety concern as they promote survival of *C. perfringens* within and outside hosts and enhance pathogenicity [29]. *C. perfringens* participate in both mono- and multi-species biofilms with enhanced survivability in the presence of oxygen [12]. In the present study, biofilm formation was confirmed in as many as 68.25% of *C. perfringens* isolates, a proportion that is slightly less than previously reported [1]. Disharmony in rates of multidrug resistance and biofilm is not uncommon and also occurs in Gram-negative bacteria [30].

Close clustering of isolates from meat, intestine, and water highlighted the ability of the organism to pass along the food chain, raising concerns of cross-contamination of surfaces, as suggested [23]. Close association of resistance to lincosamides and macrolides was attributed to macrolide-lincosamide-streptogramin B resistance, often encoded by *erm(Q)* or *erm(B)* genes [10]. Increased associations between linezolid resistance and biofilm-forming ability of the *C. perfringens* isolates are a concern, as linezolid is used for methicillin-resistant *Staphylococcus aureus* and vancomycin-resistant *Enterococci* [26].

## 5. Conclusions

This study sheds light on the concerning prevalence of multidrug-resistant, biofilm-forming virulent *C. perfringens* in broiler chicken retail points in Meghalaya, a northeastern state in India dominated by tribal populations with unique food preferences. The predominance of Type A and Type C toxinotypes in retail chicken meat and intestines merits increased attention from the perspectives of food safety and public health. The alarming rate of multidrug resistance among *C. perfringens* recovered from poultry advocates for the adequate cooking of meat before consumption. Further research is warranted to investigate the mechanisms underlying multidrug resistance in *C. perfringens* emanating from the poultry value chain.

## Figures and Tables

**Figure 1 foods-12-04185-f001:**
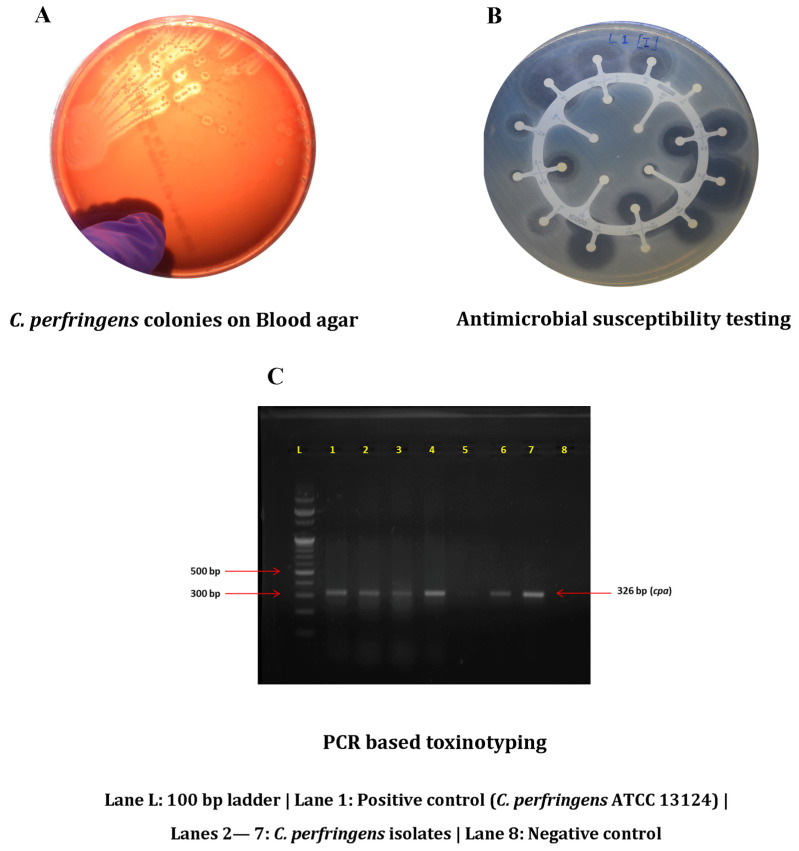
Isolation and characterization of *C. perfringens* recovered in the present study. (**A**) Isolation; (**B**) Antibiogram; (**C**) Toxinotyping.

**Figure 2 foods-12-04185-f002:**
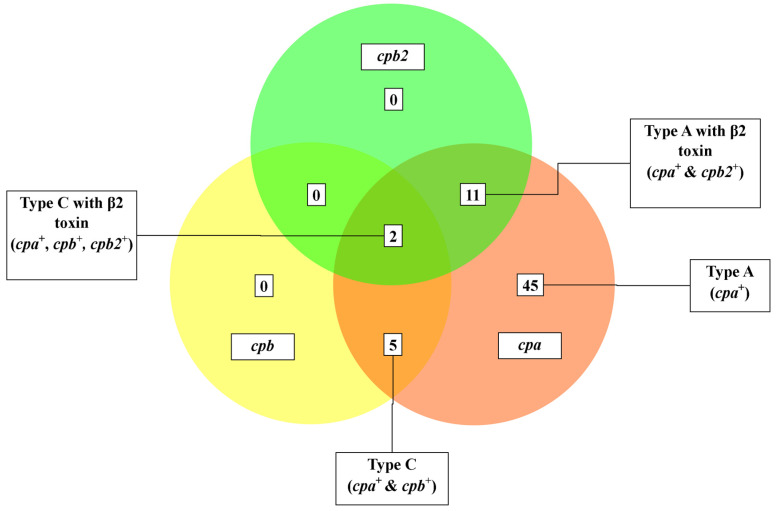
Venn diagram depicting various toxinotypes of *C. perfringens* identified in the present study.

**Figure 3 foods-12-04185-f003:**
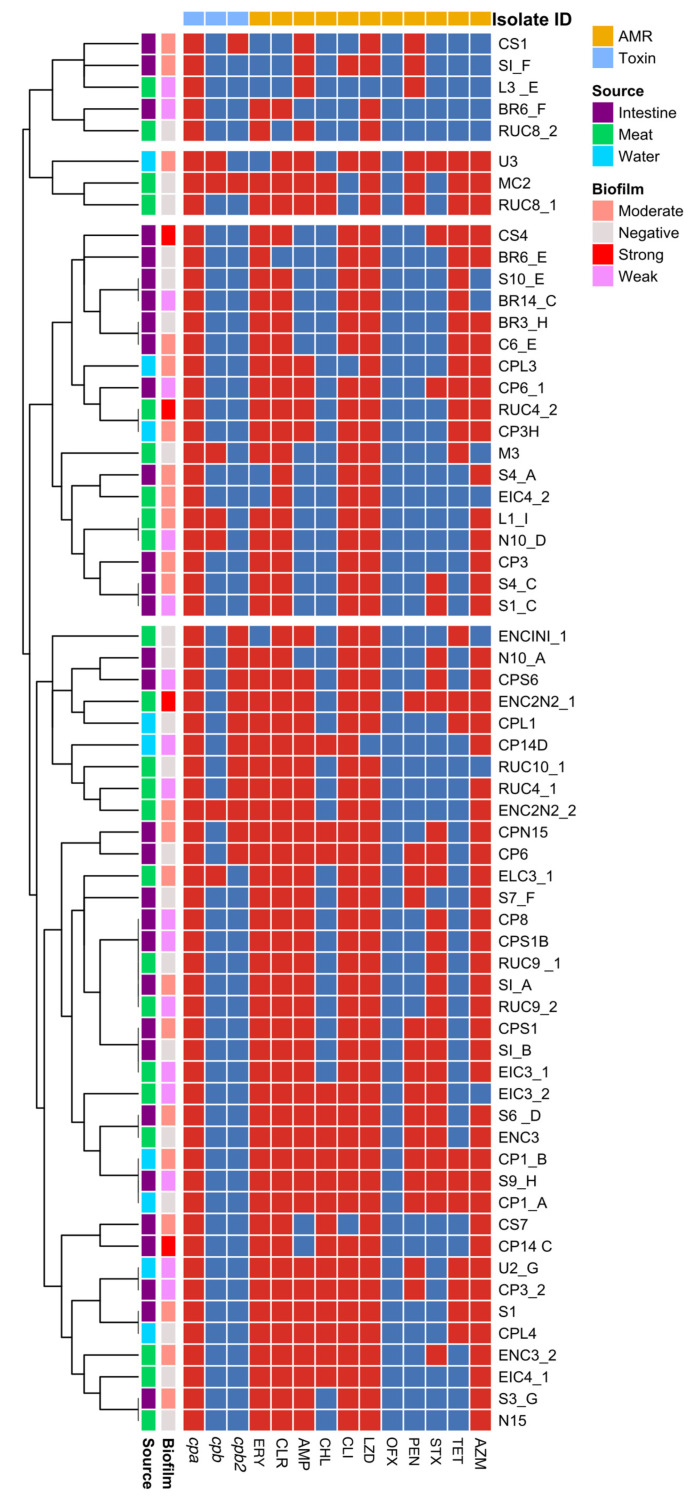
Heatmap analysis with hierarchical clustering depicting occurrence of toxin genes and resistance patterns of *C. perfringens* with respect to isolation source and biofilm-forming ability (ERY: Erythromycin; CLR: Clarithromycin; AMP: Ampicillin; CHL: Chloramphenicol; CLI: Clindamycin; LZD: Linezolid; OFX: Ofloxacin; PEN: Penicillin; STX: Co-trimoxazole; TET: Tetracycline; AZM: Azithromycin).

**Figure 4 foods-12-04185-f004:**
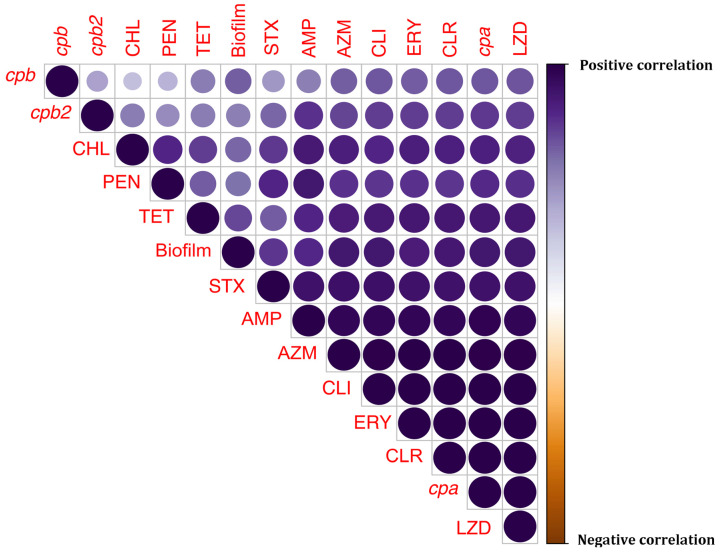
Correlation plot analysis depicting high levels of positive correlations between various attributes of *C. perfringens* (ERY: Erythromycin; CLR: Clarithromycin; AMP: Ampicillin; CHL: Chloramphenicol; CLI: Clindamycin; LZD: Linezolid; OFX: Ofloxacin; PEN: Penicillin; STX: Co-trimoxazole; TET: Tetracycline; AZM: Azithromycin).

**Table 1 foods-12-04185-t001:** Virulence repertoire and antibiogram profile of *C. perfringens* isolated from broiler value chain.

Isolate ID	Source	Toxin Genes			Designated Toxinotype	Antimicrobial Susceptibility #											MAR Index	Biofilm-Forming Ability ##
		*cpa*	*cpb*	*cpb2*		ERY	CLR	AMP	CHL	CLI	LZD	OFX	PEN	STX	TET	AZM		
SI_F	Intestine	+	−	−	Type A	I	I	R	I	R	R	S	R	S	I	S	0.36	++
SI_A	Intestine	+	−	−	Type A	R	R	R	I	R	R	S	S	R	S	R	0.64	++
SI_B	Intestine	+	−	−	Type A	R	R	R	S	R	R	S	R	R	I	R	0.73	−
S4_C	Intestine	+	−	−	Type A	R	R	S	S	R	R	S	S	R	I	R	0.55	++
S6 _D	Intestine	+	−	−	Type A	R	R	R	R	R	R	S	R	R	S	R	0.82	++
S9_H	Intestine	+	−	−	Type A	R	R	R	R	R	R	I	R	R	R	R	0.91	+
S4_A	Intestine	+	−	−	Type A	I	R	S	S	R	R	S	S	S	S	R	0.36	++
S3_G	Intestine	+	−	−	Type A	R	R	R	S	R	R	S	S	S	I	R	0.55	++
S1_C	Intestine	+	−	−	Type A	R	R	S	S	R	R	S	S	R	I	R	0.55	+
S10_E	Intestine	+	−	−	Type A	R	R	S	S	R	R	S	S	S	R	I	0.45	−
S7_F	Intestine	+	−	−	Type A	R	R	R	S	R	R	S	R	S	I	R	0.64	−
BR6_E	Intestine	+	−	−	Type A	R	S	S	S	R	R	S	S	S	R	R	0.45	−
BR6_F	Intestine	+	−	−	Type A	R	R	S	S	I	R	S	S	S	I	I	0.27	+
BR14_C	Intestine	+	−	−	Type A	R	R	S	S	R	R	S	S	S	R	I	0.45	+
CPN15	Intestine	+	−	+	Type A with *cpb2*	R	R	R	R	R	R	S	S	R	S	R	0.72	++
CP3	Intestine	+	−	−	Type A	R	R	S	S	R	R	S	S	S	I	R	0.45	++
BR3_H	Intestine	+	−	−	Type A	R	R	S	S	R	R	S	S	S	R	R	0.55	−
S1	Intestine	+	−	−	Type A	R	R	R	R	R	R	S	S	S	R	R	0.72	++
CP6	Intestine	+	−	+	Type A with *cpb2*	R	R	R	R	R	R	S	R	R	I	R	0.82	−
CS1	Intestine	+	−	+	Type A with *cpb2*	I	I	R	I	I	R	S	R	S	S	I	0.27	++
CPS1	Intestine	+	−	−	Type A	R	R	R	S	R	R	S	R	R	I	R	0.73	++
CS4	Intestine	+	−	−	Type A	R	R	S	S	R	R	S	S	R	R	R	0.64	+++
N10_A	Intestine	+	−	+	Type A with *cpb2*	R	R	S	S	R	R	S	S	R	I	R	0.55	−
C6_E	Intestine	+	−	−	Type A	R	R	S	S	R	R	S	S	S	R	R	0.55	++
CS7	Intestine	+	−	−	Type A	R	R	S	R	S	R	S	S	S	I	R	0.45	++
CP14 C	Intestine	+	−	−	Type A	R	R	S	R	R	R	S	S	S	I	R	0.55	+++
CPS6	Intestine	+	−	+	Type A with *cpb2*	R	R	R	S	R	R	S	S	R	I	R	0.64	+
CPS1B	Intestine	+	−	−	Type A	R	R	R	S	R	R	S	S	R	I	R	0.64	+
CP8	Intestine	+	−	−	Type A	R	R	R	S	R	R	S	S	R	I	R	0.64	+
CP6_1	Intestine	+	−	−	Type A	R	R	R	S	R	R	S	S	R	R	R	0.73	+
CP3_2	Intestine	+	−	−	Type A	R	R	R	R	R	R	S	R	S	R	R	0.82	+
RUC9_2	Meat	+	−	−	Type A	R	R	R	I	R	R	S	S	R	S	R	0.64	+
ENC2N2_1	Meat	+	−	+	Type A with *cpb2*	R	R	R	S	R	R	S	R	R	R	R	0.82	+++
EIC4_1	Meat	+	−	−	Type A	R	R	R	R	R	R	S	S	S	I	R	0.64	−
RUC4_1	Meat	+	−	+	Type A with *cpb2*	R	R	R	S	R	R	S	S	S	S	R	0.55	+
RUC9 _1	Meat	+	−	−	Type A	R	R	R	I	R	R	S	S	R	S	R	0.64	−
L3 _E	Meat	+	−	−	Type A	S	S	R	S	S	S	S	R	S	S	S	0.18	+
L1_I	Meat	+	+	−	Type C	R	R	S	S	R	R	S	S	S	S	R	0.45	++
MC2	Meat	+	+	+	Type C with *cpb2*	R	R	R	R	S	R	S	R	S	R	R	0.73	−
N10_D	Meat	+	+	−	Type C	R	R	S	S	R	R	S	S	S	S	R	0.45	+
EIC4_2	Meat	+	−	−	Type A	I	R	S	S	R	R	S	S	S	I	I	0.27	++
ELC3_1	Meat	+	+	−	Type C	R	R	R	I	R	R	S	R	R	I	R	0.73	++
RUC4_2	Meat	+	−	−	Type A	R	R	R	I	R	R	S	S	S	R	R	0.64	+++
ENCINI_1	Meat	+	−	+	Type A with *cpb2*	I	R	R	S	R	R	S	S	S	R	S	0.45	−
EIC3_1	Meat	+	−	−	Type A	R	R	R	S	R	R	S	R	R	I	R	0.72	+
ENC2N2_2	Meat	+	+	+	Type C with *cpb2*	R	R	R	S	R	R	S	S	S	S	R	0.55	++
EIC3_2	Meat	+	−	−	Type A	R	R	R	R	R	R	S	R	R	I	S	0.72	+
ENC3	Meat	+	−	−	Type A	R	R	R	R	R	R	S	R	R	I	R	0.82	−
M3	Meat	+	+	−	Type C	R	R	S	S	R	R	S	S	S	R	S	0.45	−
N15	Meat	+	−	−	Type A	R	R	R	S	R	R	S	S	S	S	R	0.55	−
ENC3_2	Meat	+	−	−	Type A	R	R	R	R	R	R	S	S	R	I	R	0.73	++
RUC8_2	Meat	+	−	−	Type A	R	S	R	S	I	R	S	S	S	I	S	0.27	−
RUC10_1	Meat	+	−	+	Type A with *cpb2*	R	R	R	I	R	R	S	S	S	I	S	0.45	−
RUC8_1	Meat	+	−	−	Type A	R	R	R	R	I	R	S	R	I	R	R	0.73	−
CPL4	Water	+	−	−	Type A	R	R	R	R	R	R	S	S	S	R	R	0.73	−
U2_G	Water	+	−	−	Type A	R	R	R	R	R	R	S	R	S	R	R	0.82	+
CP1_A	Water	+	−	−	Type A	R	R	R	R	R	R	S	R	R	R	R	0.91	−
CP1_B	Water	+	−	−	Type A	R	R	R	R	R	R	S	R	R	R	R	0.91	++
U3	Water	+	+	−	Type C	I	R	R	S	R	R	S	R	R	R	R	0.73	++
CP3H	Water	+	−	−	Type A	R	R	R	S	R	R	S	S	S	R	R	0.64	++
CP14D	Water	+	−	+	Type A with *cpb2*	R	R	R	R	R	S	S	S	S	I	R	0.55	+
CPL1	Water	+	−	+	Type A with *cpb2*	R	R	R	S	R	R	S	S	S	R	R	0.64	−
CPL3	Water	+	−	−	Type A	R	R	R	S	I	R	S	S	S	R	R	0.55	++

# ERY: Erythromycin; CLR: Clarithromycin; AMP: Ampicillin; CHL: Chloramphenicol; CLI: Clindamycin; LZD: Linezolid; OFX: Ofloxacin; PEN: Penicillin; STX: Co-trimoxazole; TET: Tetracycline; AZM: Azithromycin; S: Sensitive; I: Intermediate; R: Resistance. ## +: weak positive; ++: moderate positive; +++: strong positive; −: negative.

## Data Availability

All data are available from the corresponding author and will be provided on request.

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
