# Peer review of "High Prevalence of Multidrug-Resistant, Biofilm-Forming Virulent Clostridium perfringens in Broiler Chicken Retail Points in Northeast India"

_foods, 2023, doi:10.3390/foods12224185_

Round 1
Reviewer 1 Report
Comments and Suggestions for Authors
The manuscript presents the results on the presence and characterization of food-borne pathogen C. perfringens in broiler chicken value chain in a northeast Indian state. This region is especially interesting due to its dominant tribal population with higher interest in consuming chicken intestines. Therefore, determining the prevalence, virulence and antimicrobial resistance of C. perfringens provides a valuable insight from the food-safety aspect.
First line of the Abstract needs to be rewritten.
Figure 2. is confusing and does not represent the results given in the result section properly. The Venn diagram should be more clearly presented with detailed description in figure caption.
The antibiotics being tested should be assigned with their appropriate abbreviations listed in the results.
Some order in Table 1 would be advisable, e.g. by source or designated toxinotype and explanation for abbreviations R, S and I should be given in table description.
Comments on the Quality of English Language
Some sentences need to be revised, some words (verbs or prepositions) are often missing which affects the meaning, e.g. first sentence in paragraph 3.3.
Author Response
First line of the Abstract needs to be rewritten.
Reply: Rewritten as “In light of the significant public health and food safety implications associated with Clostridium perfringens, this study aimed to isolate and characterize C. perfringens in samples obtained from broiler chicken retail locations in Meghalaya, Northeastern India.”
Figure 2. is confusing and does not represent the results given in the result section properly. The Venn diagram should be more clearly presented with detailed description in figure caption.
Reply: Thanks to the suggestion of the learned reviewer, we have replaced Figure 2 with a new figure that presents the data more clearly
The antibiotics being tested should be assigned with their appropriate abbreviations listed in the results.
Reply: In the Materials and Methods section (section 2.3) the abbreviations of the antibiotics have been included as follows: Erythromycin (ERY, 15 µg), clarithromycin (CLR, 15 µg), ampicillin (AMP, 10 µg), chloramphenicol (CHL, 30 µg), clindamycin (CLI, 2 µg), linezolid (LZD, 30 µg), ofloxacin (OFX, 5 µg), penicillin (PEN, 10 U), co-trimoxazole (STX, 25 µg), tetracycline (TET, 30 µg) and azithromycin (AZM, 15 µg).
Some order in Table 1 would be advisable, e.g., by source or designated toxinotype and explanation for abbreviations R, S and I should be given in table description.
Reply: Thank you for the suggestion. We have implemented the corrections (source-wise), greatly enhancing clarity for readers.
Some sentences need to be revised, some words (verbs or prepositions) are often missing which affects the meaning, e.g., first sentence in paragraph 3.3.
Reply: Made necessary corrections as suggested.
Reviewer 2 Report
Comments and Suggestions for Authors
The article entitled "High prevalence of multidrug-resistant, biofilm forming viru-lent Clostridium perfringens in broiler chicken retail points in Northeast India" is a local good manuscript which is adequated for this journal. However, I recommend that authors should read the instructions for authors due to that references in reference section is uncorrect according to Foods-MDPI
Comments on the Quality of English Language
The article entitled "High prevalence of multidrug-resistant, biofilm forming viru-lent Clostridium perfringens in broiler chicken retail points in Northeast India" is a local good manuscript which is adequated for this journal. However, I recommend that authors should read the instructions for authors due to that references in reference section is uncorrect according to Foods-MDPI
Author Response
The article entitled "High prevalence of multidrug-resistant, biofilm forming viru-lent Clostridium perfringens in broiler chicken retail points in Northeast India" is a local good manuscript which is adequated for this journal. However, I recommend that authors should read the instructions for authors due to those references in reference section is incorrect according to Foods-MDPI
Reply: Corrected following the guidelines of Foods-MDPI.
Reviewer 3 Report
Comments and Suggestions for Authors
The main objective of this study was to determine the food-borne threat posed by C. perfringens.
The study is well written and some minor corrections are needed as follows:
In table 1 can not be read all columns of "Antimicrobial Susceptibility"
Figure 4- does the colour and the size change, and indicate the "pozitive" correlations?
It is not clear for what was the ANOVA used, and it is not mentioned which tools were used for the presented charts (figures 3 and 4) because this is not available in the MS Excel.
Sincerly
Author Response
The main objective of this study was to determine the food-borne threat posed by C. perfringens. The study is well written and some minor corrections are needed as follows:
In table 1 cannot be read all columns of "Antimicrobial Susceptibility"
Reply: Corrected as suggested
Figure 4- does the colour and the size change, and indicate the "positive" correlations?
Reply: In correlation plot, the colour and size of the circles represent the magnitude and direction of correlation. In the given figure, all parameters included indicated positive correlations. (Included in the text)
It is not clear for what was the ANOVA used, and it is not mentioned which tools were used for the presented charts (figures 3 and 4) because this is not available in the MS Excel.
Reply: ANOVA was a typographical error. The tools used for constructing figures 3 and 4 are given under the heading 2.5 of Materials and Methods section.
Reviewer 4 Report
Comments and Suggestions for Authors
Dear Aurhors,
Dear Editor,
Problem with Clostridium perfringens is very important topic for food technology. Addressing these issues requires a comprehensive approach involving surveillance, regulatory measures, and public awareness. This article's findings could contribute to the development of strategies to mitigate the risks associated with the presence of multidrug-resistant and virulent Clostridium perfringens in the food supply chain, particularly in Northeast India.
Intoduction is well written. Necessary information introducing the subject of research is included.
Material and methods is well written. Please give desription to "PBS". The value of sample is enought. Please explain how the samples were collected - e.g. from butchers (during work or after washing hands), slaughterwater (or after washing the meat?).
The overall occurrence of C. perfringens in retail chicken associated samples was 22.5% (63/280, 17.74 – 27.85, 95%CI) - please add 63/280 carcasses/chicken/samples; what is the value 17.74-27.85? and why is the value of 95% Cl added? I don't think it's necessary if you write about it in the methodology once.
In Figure 2. You have Type A value 49, and I think it should be 45, because You have information: "predominance of Type A toxinotype (45/63, 71.43%; 58.65 – 82.11, 95% CI)". And Authors have in Figure 2. informations for cpb "0" and for cpb2 also "0", and in text we have informations: "cpb2 (13/63...)" & "cpb (7/63...)". Or I understand this wrong?
Please give informations what is this: E-CLR-AMP-CD-LZ-COT-AZM, E-CLR-AMP-CD-LZ-AZM, E-CLR-AMP-CD-LZ-P-COT-AZM.
Please give the explain to "Isolate S9_H from chicken intestine and
isolates CP1_A and CP1_B recovered from water samples" and other name from Table 1.
Please describe the schema in page 8. The title and abbreviations.
Discussion:
Do You think "in only 68.25% of ....." is only or in as many as 68.25% ? I think near 70% is a big value.
Conclusions: Please summarize why the study's results are important for public health, food safety, and the specific geographic area, give point out areas where there is a need for further research, especially if there are unexplained aspects related to the presence of Clostridium perfringens in broiler chicken retail points in the given location and Draw attention to the alarming level of multidrug resistance in Clostridium perfringens isolates and underscore the necessity of adequately cooking meat before consumption to minimize the risk associated with the presence of these bacteria.
Good luck!
Comments on the Quality of English Language
Minor editing of English language required.
Author Response
Material and methods is well written. Please give description to "PBS". The value of sample is enough. Please explain how the samples were collected - e.g., from butchers (during work or after washing hands), slaughterwater (or after washing the meat?).
Reply: Expanded PBS. Hand swabs were collected during meat cutting, and slaughter water refers to the water used for slaughtering and meat cutting at the retail points.
The overall occurrence of C. perfringens in retail chicken associated samples was 22.5% (63/280, 17.74 – 27.85, 95% CI) - please add 63/280 carcasses/chicken/samples; what is the value 17.74-27.85? and why is the value of 95% Cl added? I don't think it's necessary if you write about it in the methodology once.
Reply: For all the proportions, confidence intervals (CI) were calculated with 95% confidence level using binomial exact calculation. For better clarity, we have edited like this 22.5% (17.74 – 27.85, 95% CI; 63/280).
In Figure 2. You have Type A value 49, and I think it should be 45, because You have information: "predominance of Type A toxinotype (45/63, 71.43%; 58.65 – 82.11, 95% CI)". And Authors have in Figure 2. informations for cpb "0" and for cpb2 also "0", and in text we have informations: "cpb2 (13/63...)" & "cpb (7/63...)". Or I understand this wrong?
Reply: We have replaced Figure 2 with a new figure that presents the data more clearly
Please give informations what is this: E-CLR-AMP-CD-LZ-COT-AZM, E-CLR-AMP-CD-LZ-AZM, E-CLR-AMP-CD-LZ-P-COT-AZM.
Reply: These are antibiotic resistance profiles for different MDR isolates. Each combination of letters corresponds to the resistance of the organism to specific antibiotics. For example, in the first pattern "ERY-CLR-AMP-CLI-LZD-STX-AZM," it suggests resistance to Erythromycin, Clarithromycin, Ampicillin, Clindamycin, Linezolid, Cotrimoxazole, and Azithromycin.
Please give the explain to "Isolate S9_H from chicken intestine and
isolates CP1_A and CP1_B recovered from water samples" and other name from Table 1.
Reply: These are isolate IDs used in our laboratory.
Please describe the schema in page 8. The title and abbreviations.
Abbreviated as suggested
Reply:
Discussion:
Do You think "in only 68.25% of ....." is only or in as many as 68.25% ? I think near 70% is a big value.
Reply: "Yes, as rightly pointed out by the learned reviewer, it has significant value. However, in comparison to previous studies that reported around 85% (Milton et al., 2022; Charlebois et al., 2016), our findings indicated a slightly lower value. We have revised the statement in accordance with the suggestions provided by the reviewer.
Conclusions: Please summarize why the study's results are important for public health, food safety, and the specific geographic area, give point out areas where there is a need for further research, especially if there are unexplained aspects related to the presence of Clostridium perfringens in broiler chicken retail points in the given location and Draw attention to the alarming level of multidrug resistance in Clostridium perfringens isolates and underscore the necessity of adequately cooking meat before consumption to minimize the risk associated with the presence of these bacteria.
Reply: Modified conclusions: This study sheds light on the concerning prevalence of multidrug-resistant, biofilm forming virulent C. perfringens in broiler chicken retail points in Meghalaya, a Northeastern state of India dominated by tribal population with unique food preferences. The predominance of Type A and Type C toxinotypes in retail chicken meat and intestines merits increased attention from the perspectives of food safety and public health. The alarming rate of multi-drug resistance among C. perfringens recovered from poultry advocates for adequate cooking of meat before consumption. Further research is warranted to investigate the mechanisms underlying multidrug resistance in C. perfringens emanating from the poultry value chain.